# AI-Based Approach to One-Click Chronic Subdural Hematoma Segmentation Using Computed Tomography Images

**DOI:** 10.3390/s24030721

**Published:** 2024-01-23

**Authors:** Andrey Petrov, Alexey Kashevnik, Mikhail Haleev, Ammar Ali, Arkady Ivanov, Konstantin Samochernykh, Larisa Rozhchenko, Vasiliy Bobinov

**Affiliations:** 1Polenov Russian Research Institute of Neurosurgery, Almazov National Medical Research Center, 191014 St. Petersburg, Russia; doctorpetrovandrey@gmail.com (A.P.); arkady.neuro@gmail.com (A.I.); samochernykh_ka@almazovcentre.ru (K.S.); rozhch@mail.ru (L.R.); neyro.bobinov@yandex.ru (V.B.); 2St. Petersburg Federal Research Center of the Russian Academy of Sciences (SPC RAS), 199178 St. Petersburg, Russia; haleev.m@iias.spb.su; 3Information Technologies and Programming Faculty, ITMO University, 197101 St. Petersburg, Russia; ammarali32@itmo.ru

**Keywords:** hematoma segmentation, computed tomography, computer vision

## Abstract

This paper presents a computer vision-based approach to chronic subdural hematoma segmentation that can be performed by one click. Chronic subdural hematoma is estimated to occur in 0.002–0.02% of the general population each year and the risk increases with age, with a high frequency of about 0.05–0.06% in people aged 70 years and above. In our research, we developed our own dataset, which includes 53 series of CT scans collected from 21 patients with one or two hematomas. Based on the dataset, we trained two neural network models based on U-Net architecture to automate the manual segmentation process. One of the models performed segmentation based only on the current frame, while the other additionally processed multiple adjacent images to provide context, a technique that is more similar to the behavior of a doctor. We used a 10-fold cross-validation technique to better estimate the developed models’ efficiency. We used the Dice metric for segmentation accuracy estimation, which was 0.77. Also, for testing our approach, we used scans from five additional patients who did not form part of the dataset, and created a scenario in which three medical experts carried out a hematoma segmentation before we carried out segmentation using our best model. We developed the OsiriX DICOM Viewer plugin to implement our solution into the segmentation process. We compared the segmentation time, which was more than seven times faster using the one-click approach, and the experts agreed that the segmentation quality was acceptable for clinical usage.

## 1. Introduction

Chronic subdural hematoma (CSDH) is a type of human brain hematoma whose clinical manifestations can be seen three weeks after the event that caused them [1,2,3,4]. CSDHs are very common (0.002–0.02%) and generally occur in elderly people (0.05–0.06% of people aged 70 years and above). The authors of paper [5] note that CSDHs are among the most common diseases in neurosurgery. For the diagnosis and treatment of chronic subdural hematomas, volumetric analysis is usually used [6]. Accurately measuring the volume of a CSDH is a crucial indicator for assessing the need for surgery and monitoring disease progression [7]. These measurements also serve as objective indicators for evaluating the effectiveness of treatment. Most clinicians use the Coniglobus formula, which calculates the hematoma volume as length cubed to determine the hematoma measurement [8,9]. Although the technique is fairly accurate for measuring ellipsoidal hematomas with small amounts of bleeding, it is not precise enough when dealing with cases of intraventricular hemorrhage and subdural hematomas with more significant bleeding and irregular shapes [10].

Since Lee K. S. et al.’s [11] initial publication in 2001, several authors have proposed the impact of “skull shape”—whether symmetrical or asymmetrical—on the localization of the CSDH [12,13]. Hsieh and colleagues [12] employed a semi-automated technique, following the concept detailed by Zonenshayn in 2004 [14], to gauge plagiocephaly. Computed tomography (CT) images were imported into a LabVIEW software system (National Instruments, Austin, TX, USA). The midline comprised the nasion and inion, which were marked manually using black dots. The study was further automated to encompass cranial bone subtraction, pixel numbering and identification of each cranial region. The axis was established along the midline and flipping one side of the skull horizontally to the other defined the overlapping and non-overlapping regions. The non-overlapping side with the largest area was considered dominant [12].

In this paper, we present an approach to one-click chronic subdural hematoma segmentation based on computer vision analysis of computed tomography images. To carry out this method, we propose and comprehensively compare two neural network models for hematoma segmentation and integrate the developed models into the OsiriX DICOM Viewer, which allows medical expert to use the models as a one-click solution. The first model is based on the well-known U-Net architecture (which is widely used for medical data segmentation) and generates results based on consideration of one data slice, while the second model considers several slices (contextual information) for results generation (2.5D model). To train the models, we generated a dataset that contains 53 series of CT scans collected from 21 patients with one or two hematomas. Based on this dataset, we trained the proposed models and calculated the Dice metric. For the 2.5D model, we obtained a result of 0.77 (Dice metric) calculated with a 10-fold cross-validation scenario. Then, we estimated the model in a real medical scenario. Three medical experts participated in the experiment and CT images of five additional patients were considered. We used two scenarios: a manual one, in which the medical experts implemented the hematoma segmentation, and a one-click scenario with the proposed 2.5D model. We compared the segmentation time, which was more than seven times faster using the one-click approach, and the experts agreed that the segmentation quality was acceptable for clinical usage.

The scientific novelty of the paper is that it offers the following: (1) a unique dataset of segmented chronic subdural hematomas; (2) an innovative approach to one-click chronic subdural hematoma segmentation based on the OsiriX DICOM Viewer; and (3) a proposed effective neural network 2.5D model for chronic subdural hematoma segmentation.

The rest of the paper is structured as follows: Section 2 gives an overview of the related research on the topic, Section 3 describes the approach used, including the acquired dataset description and the proposed neural network models, Section 4 presents the results, including the quantitative and qualitative results, and Section 5 summarizes the paper.

## 2. Related Work

The authors of paper [15] proposed a novel approach to the neural segmentation of traumatic brain injury (TBI) hematomas. The approach was explored using a multi-view convolutional neural network (CNN). The study utilized a dataset comprising 828 patients with scans collected within a 24 h window post-injury. The authors quantified the efficacy of this approach using the Dice similarity coefficient, a spatial statistic for gauging image segmentation performance, and the model achieved a score of up to 0.669. In addition to the primary focus on hematoma segmentation, the study also constructed two baseline models for predicting six-month mortality post-TBI. The first baseline model integrated International Mission for Prognosis and Analysis of Clinical Trials (IMPACT) models with logistic regression. In contrast, the second baseline model employed a random forest algorithm, a robust machine learning technique known for its interpretability and handling of high-dimensional data. The authors employed a rigorous evaluation methodology to ensure the reliability and robustness of these predictive models. This involved performing 10-fold cross-validation, a resampling procedure used to evaluate machine learning models on a limited data sample, repeated 50 times. This repetition aids in mitigating any bias arising from the arbitrary division of training and test data, thereby providing a more accurate estimate of model performance. The authors evaluated the predictive power of hematoma-relevant features, providing valuable insights into the role and significance of these features in predicting patient outcomes post-TBI.

In the scientific study conducted by the authors of [16], a sophisticated approach to the neural segmentation of subdural hematoma was explored using a 3D convolutional neural network (CNN). The study utilized a dataset comprising 128 scans. The efficacy of this approach was quantified using the Dice similarity coefficient, a spatial statistic used to gauge the performance of image segmentation, with the model achieving a score of up to 0.806. This high score indicates a strong agreement between the predicted segmentation and the ground truth, demonstrating the effectiveness of using 3D CNN for subdural hematoma segmentation.

The authors of paper [17] utilized a convolutional neural network for intracerebral hemorrhage segmentation. They achieved a Dice similarity coefficient of up to 0.75, indicating effective segmentation. The dataset had 121 scans taken within 4 days of patient admission to the local hospital with intracerebral hemorrhages. The model followed U-Net architecture with periodic batch normalization and five paired convolution–deconvolution layers.

The authors of paper [18] developed a multiresolution binary level set method for hematoma segmentation, achieving a Dice overlap metric of up to 0.8. This method modifies the Song–Chan algorithm to compute the edge length in irregular image margins, making it suitable for segmenting objects in any position, especially near the image margins. The authors utilized image pyramids for multiresolution processing, allowing the binary level set method to work on images with a reduced resolution and size. This approach improves efficiency by processing a point on a lower-resolution image instead of a block or a strip at the original resolution. The method was successfully applied to segment intracranial hematomas on brain CT slices, including epidural and subdural hematomas. The segmentation results were comparable to those achieved by human experts and were obtained in seconds.

In the paper [19], the authors developed a U-Net-based model for segmenting tiny intracerebral hemorrhage regions in brain CT images. The model was trained on a dataset of 318 intracerebral hemorrhage images. To handle the tiny intracerebral hemorrhage regions, the authors proposed a residual hybrid atrous convolution strategy and introduced a multi-object function for joint optimization. The model outperformed previous methods in both internal testing data and external clinical data, indicating its potential for clinical application. The performance of the developed segmentation algorithm was analyzed using the Dice coefficient and Jaccard index. The proposed model outperformed previous methods and achieved satisfying segmentation performance for intracranial hemorrhage testing in both the internal testing data (Dice 0.725, Jaccard 0.605) and the external clinical data (Dice 0.869, Jaccard 0.774), indicating its superior segmentation effect and potential clinical prospects.

In paper [20], the authors developed an automatic segmentation network, CHSNet (cascaded neural network with hard example mining and semi-focal loss), for segmenting lesions in cranial CT images characteristic of acute cerebral hemorrhage. The study utilized a dataset comprising 5998 cranial CT slices collected from 203 cases and achieved a Dice similarity coefficient of up to 0.918, indicating effective segmentation. The network architecture includes an encoding–decoding backbone, Res-RCL module, Atrous Spatial Pyramid Pooling and Attention Gate. These components enhance the feature representation of high-density regions and capture multi-scale and up–down information on the target location.

The authors of paper [21] developed an AMD-DAS (domain adaptation segmentation model that can be trained across modalities and diseases), which is an unsupervised domain adaptation method for the neural segmentation of brain CT scans using CycleGAN (deep-learning architecture using two generative adversarial networks). This method improves the diagnosis of brain diseases by automatically segmenting hemorrhagic lesions. It involves a two-stage process: training a pseudo-CT image synthesis network and then using these images to train a domain adaptation segmentation model. The method outperformed the basic one-stage domain adaptation segmentation method by a +11.55 Dice score, achieving an 86.93 Dice score. Notably, it can be trained without ground truth labels, increasing its potential for clinical application. The training set included 484 scans. Each scan contained 155 images.

The authors of paper [22] developed a method for segmenting hemorrhage subtypes in traumatic brain injury patients using EfficientNet B4 and DeepMedic v0.8.4. The method uses CT scans with bone window as input and includes a post-processing step for refinement. The dataset comprised CT scans from 153 patients. The method achieved median Dice similarity coefficients higher than 0.37 for each hemorrhage subtype, demonstrating its effectiveness.

The authors of paper [23] improved the U-Net model for better segmentation of irregular intracranial hemorrhage lesions in CT images. They introduced a residual octave convolution module and a mixed attention mechanism. The dataset included CT scans from 40 patients. The improved U-Net demonstrated higher accuracy, improving the Dice coefficient from 0.74 to 0.86, outperforming both the original U-Net and the region growing algorithm.

The authors of paper [24] developed a method for segmentation with the energy function of cerebral hemorrhages in CT images, considering the irregularity and variability of blood clots. The method combines shape and area constraints and was tested on 42 patients, achieving a Dice score of over 0.93. They used Zernike moment-based extraction of CT images.

The articles reviewed lead to several key observations. Firstly, the subject matter is of high relevance and has been extensively studied. However, there is a scarcity of research, specifically on subdural hematomas, which can be attributed to the complexity of segmentation, with only a few works such as [16,18] addressing this area. In addition, U-Net architecture is predominantly used in these studies [17,19,20,23], while CycleGAN [21], EfficientNet and DeepMedic [22] are also employed. Some studies have also explored the approach to recognize hematomas [18,24]. Lastly, overlap metrics, namely, the Jaccard and Dice metrics, are commonly used for model evaluation.

## 3. Approach

### 3.1. General Scheme

The automated hematoma segmentation process is designed to detect and segment hematoma regions in head computed tomography (CT) images of patients with subdural hematomas (see Figure 1).

We considered two scenarios for hematoma segmentation: a manual scenario and an automated scenario. Manual hematoma segmentation consists of the following steps:Brain CT visualization: This is the initial step, in which the brain CT scans are visualized using a specialized software program. This allows the human operator to view the scans in detail;Manual hematoma segmentation: In this step, a trained professional manually identifies and segments the hematoma from the CT scan. This process requires a high level of expertise and can be time-consuming;Hematoma metric calculation: After the segmentation, various metrics related to the hematoma, such as its volume, location and shape, are calculated. These metrics are crucial for further diagnosis and treatment planning.

The automated hematoma segmentation scenario includes the following steps:Brain CT visualization: Like the manual process, the brain CT scans are visualized using a specialized software program;Segmentation system launch: In this step, an automated segmentation system is launched with one click. This system uses machine learning algorithms to identify and segment the hematoma;Automated hematoma segmentation: The system automatically segments the hematoma from the CT scan. This process is typically faster and less prone to human error compared to manual segmentation;Human-based verification of hematoma segmentation: Despite the automation, a human operator verifies the segmentation results to ensure accuracy. This step acts as a safeguard against any potential errors by the automated system;Hematoma metric calculation: Like the manual process, various hematoma metrics are calculated after segmentation.

The main differences between these two scenarios lie in steps 2 (manual scenario) and 3 (automated scenario). In the manual scenario, a human operator does all the work, which can be time-consuming and prone to errors. On the other hand, in the automated scenario, a machine learning system does most of the work, which is typically faster and more accurate. However, even in the automated scenario, human verification is still necessary to ensure accuracy.

### 3.2. Dataset

The data were collected at the Almazov National Research Medical Centre (non-randomized monocentric case–control study). Patients were enrolled between November 2021 and May 2023. Approval for the study was granted and monitored by the local ethics committee of the National Research Medical Center, Almazov. The research was conducted in accordance with the 2013 version of the Helsinki Declaration by the World Medical Association. The Ethics Committee of the National Medical Research Centre, Almazov, approved the study (12 July 2019) and the approval code is 07-19. All enrolled patients agreed in written form to the anonymous publication of their data.

The examinations were conducted using a Philips (USA) Ingenuity Core multispiral computed tomography scanner with 128 slices. The acquisitions were helical with a slice thickness of 1 mm, a slice spacing of 0.5 mm and a total of 302 slices. The total acquisition time was 3.429 s.

Our dataset contains 53 series of CT scans of people over 18 years old with the presence of at least one CSDH. The average age of the patients was 61 ± 14 (m ± SD). The study did not include pregnant women or patients who had blood diseases, bronchial asthma, diabetes mellitus, oncological diseases or rheumatological diseases. Also, we did not include patients taking anticoagulants or antiplatelet agents. More details about the dataset are presented in Table 1.

We determined the volume of the CSDH manually by measuring the specific volumes, defining the area of interest using OsiriX software (Osirix for Mac, version 11.0) on a pre-embolization of NCCT [2]. In cases where the hematoma was located bilaterally, the hematoma volumes were summed and a total hematoma volume was obtained.

### 3.3. Neural Network Model

We proposed a model based on U-Net architecture, which is the most prominent neural network architecture for medical image analysis. Firstly, we tried a 2D model, which offered computational efficiency by focusing on individual image slices, but we then proposed using a 2.5D model, which struck a balance between computational efficiency and contextual understanding by incorporating contextual information between CT slices. On the one hand, 2D segmentation models operate on individual image slices, making them computationally efficient but lacking in contextual information between slices. On the other hand, 2.5D segmentation models process a few consecutive slices at a time, providing some context while maintaining relative computational efficiency (see Figure 2).

#### 3.3.1. Description of 2D Model

We proposed using a U-Net model, which is a convolutional neural network designed for biomedical image segmentation, to perform the segmentation task (see Figure 3). The input images were grayscale and had a resolution of 512 × 512 pixels. Then, we implemented augmentation of the CT images to extend our dataset (see Figure 4). Next, we implemented the training process interactively until the Dice metric was improving. Once the Dice metric improvement was completed, we stopped the training. We set the learning rate to *0.0001* (found empirically) and used the EfficientNet-B4 as the backbone of the U-Net model. We used softmax as the output activation function and relu as the layer activation function. We used DiceLoss as the loss function and applied and compared different threshold values (0.3, 0.5 and 0.7) to binarize the output masks (see Section 4).

#### 3.3.2. Description of 2.5D Model

We proposed using adjacent frames of a CT image as a “context” for recognizing hematomas. For this purpose, we proposed an architecture based on U-Net architecture, which supplies several channels as inputs, each of which corresponds to a separate frame (see Figure 5). The training process is similar to the training process of the 2D model (see Figure 6). The input images were composed of 5 images, which were cropped to up to 87.5% of their original size to remove the CT image borders. We trained the model for *40* epochs (after empirical investigation, we considered that 40 were sufficient), using Adam as the optimizer with the same learning rate (*0.0001*) and a weight decay of *0.001*. We used a gamma scheduler with a factor of *0.96* to adjust the learning rate. We applied data augmentation techniques with a probability of *0.4*, including strong augmentations.

### 3.4. Cross-Validation

We proposed employing a 10-fold stratified cross-validation technique since we did not have a large amount of data. The patient dataset was divided into *10* subsets or “folds”, ensuring that each fold had an approximately equal number of hematoma cases. This method, known as stratification, is crucial when dealing with datasets where certain outcomes are overrepresented, as it maintains the ratio of the predictive classes. The cross-validation scheme consists of the following steps (see Figure 7):

Gathering data from dataset and forming a cross-validation set;Dividing the cross-validation set into *10* different folds. The “stratified” part of StratifiedKFold means that each fold will contain roughly the same proportions of the different types of class labels as the original dataset (see Table 2). We found that embolization and the number of hematomas on the CT images influenced the neural network’s ability to recognize hematomas. This is important for maintaining a representative sample of the data in each fold;Processing *i*-th fold: Selecting one fold to be the test set and the remaining folds to be the training set. The *i*-th fold refers to the current iteration of the cross-validation process;Training the model using the training set;Testing the model using the test set;Calculating performance metrics based on the model’s predictions and the actual target values;After completing steps 3–6 for all folds, calculating the final performance metrics. These are the averages of the performance metrics from each fold, which offer the possibility of a robust estimate of how well the model is likely to perform on unseen data.

We repeated steps 3–6 for each fold using the StratifiedKFold cross-validation process. Each time, we selected a different fold to be a test set and the remaining folds to be training sets.

## 4. Results

This section outlines the usage scenarios and the results of the experiments, obtained through metric calculation.

### 4.1. Usage Scenarios

The proposed usage scenario described in Section 3 for the 2D model begins with the input of a CT frame (see Figure 8). To enhance prediction accuracy, a mirrored version of the CT frame is also used. The U-Net model then predicts a recognition mask for both the original and mirrored CT frames. These recognition masks are binary images that indicate where the hematoma is present in the CT frames. The predicted masks for the original and mirrored frames are then averaged to obtain a final, more accurate recognition mask. This final mask is used for further analysis, such as calculating various metrics related to the hematoma. This approach leverages the power of the U-Net model for automated hematoma segmentation and incorporates an innovative technique using mirrored frames to improve prediction accuracy.

The proposed usage scenario described in Section 3 for the 2.5D model begins with the selection of several sequential CT frames (see Figure 9). These frames are then stacked together to form a multi-channel input image, similarly to how color images have red, green and blue channels. Each channel in this case represents a different CT frame. This multi-channel image is then fed into the U-Net model. The model predicts a single recognition mask for the multi-channel image. This recognition mask is a binary image that indicates where the hematoma is present across all the CT frames. This final mask represents the segmented hematoma across all input frames and is used for further analysis, such as calculating various metrics related to the hematoma. This approach leverages the power of the U-Net model for automated hematoma segmentation and incorporates a technique using multiple frames as input channels to provide more context to the model, thereby improving prediction accuracy.

### 4.2. Quantitative Results

In our study, we employed three distinct test sets for evaluating the 2D model with different parameters: a test series set (one series that is not included in the training dataset), a training set (one series that is included in the training dataset) and a mixed set (mixed slices from different research). For the assessment of the 2.5D model with different parameters, we implemented a cross-validation strategy with ten folds. We then computed the Jaccard and Dice metrics for each set to gauge their performance (see Table 3 and Table 4). 

Based on Table 3, it can be inferred that augmentation prevents the models from overfitting, as evidenced by the smaller difference between the performance results on the series from the training and test datasets. Furthermore, according to Table 4, cross-validation allows for more stable and justified performance evaluations of the models.

For the 2D model, the best-performing version on the first test set was the one with a threshold of 0.5 and with augmentation. However, on the second and third test sets, the best model was the one with a threshold of 0.5 and without augmentation. For the 2.5D model, the best performers were the versions with three slices and thresholds of 0.1 and 0.2.

### 4.3. Qualitative Results

For clinical acceptance, the CT scans of the brains of five patients with CSDHs were evaluated as part of the standard operating procedure. These scans were obtained prior to endovascular surgery, and they were not part of the primary dataset. The patients had a mean age of 61 ± 15 (m ± SD), and there were four males and one female among them. The CSDH was located bilaterally in two patients and unilaterally in three. Three specialist radiologists (Exp#1, Exp#2, Exp#3) with more than 10 years of experience performed the evaluations. The primary acceptance criterion was based on a time study. Time study is a technique for assessing the efficiency of software by recording and measuring the duration of tasks carried out within a workflow. In this study, it was used to measure the time taken to produce a report on manual and automatic CSDH volume measurements (including manual corrections). The results of the clinical acceptance process are shown in Table 5.

A survey of the experts was conducted, evaluating the automatic calculation’s ease of use, subjective assessment of the method accuracy and the potential for widespread clinical use. The automated series processing of all five of these patients was confirmed by the experts to be suitable for clinical usage.

## 5. Conclusions

This paper presents a computer vision-based approach to chronic subdural hematoma segmentation by using computed tomography images. The segmentation of subdural hematomas allows us to measure their volume in order to make decisions about human health and about whether or not an embolization procedure is required. We proposed two neural network models. The first model is based on U-Net architecture and analyzed each scan separately. The second proposed model extends the previous one by including analysis of a series of scans (taking context into account). We tested the proposed models on test sets from our own dataset, which contained 53 series of CT scans collected from 21 patients, as well as on 5 additional patients. We estimated our models based on the Dice metric and our experiments show the best result as 0.77. We developed the OsiriX DICOM Viewer plugin using our model, which allows us to use our solution in a specialized medical program. We conducted experiments in which three experts carried out hematoma segmentation both manually and using our developed models. The experiments showed that by using the developed models, processing time was reduced by an average of more than seven times and the experts considered that the neural network segmentation achieved was acceptable for clinical usage.

The main limitation of the paper is that we do not offer an automated approach to hematoma segmentation. We propose a one-click chronic subdural hematoma segmentation for medical professionals, which estimates results and makes human-based decisions about patient diagnosis.

## Figures and Tables

**Figure 1 sensors-24-00721-f001:**
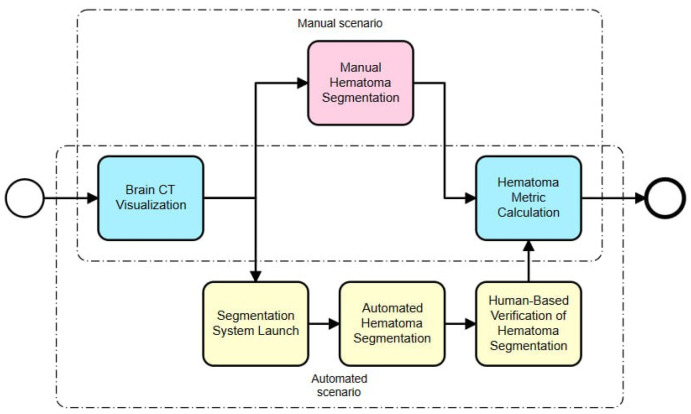
General scheme of automated hematoma segmentation process.

**Figure 2 sensors-24-00721-f002:**
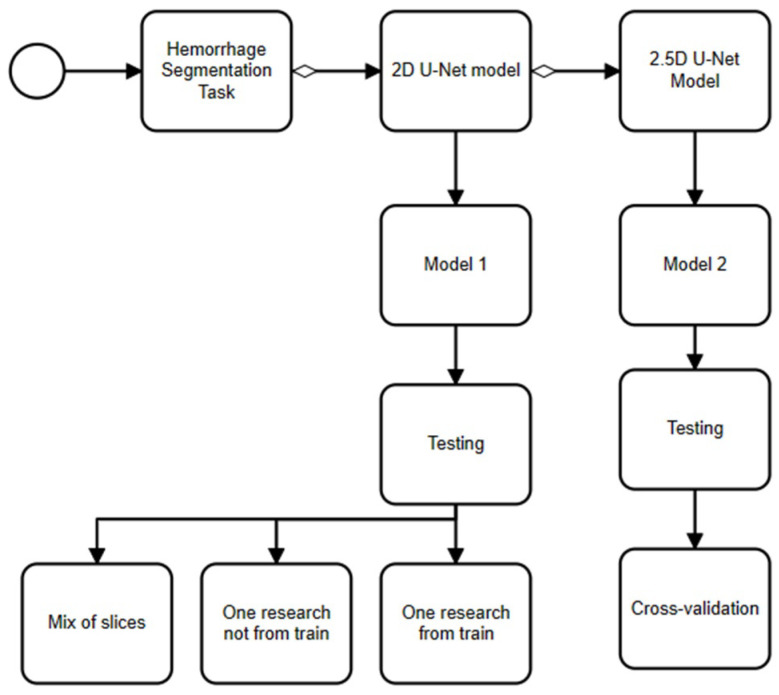
An approach to chronic subdural hematoma segmentation.

**Figure 3 sensors-24-00721-f003:**
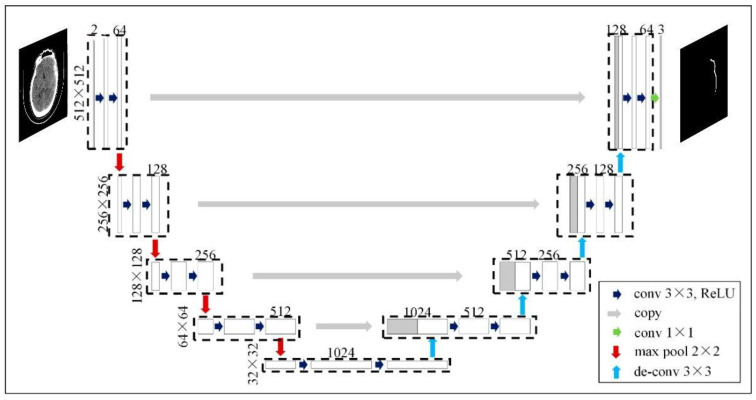
Example of U-Net architecture for input image 512 × 512 (adapted from [25]).

**Figure 4 sensors-24-00721-f004:**
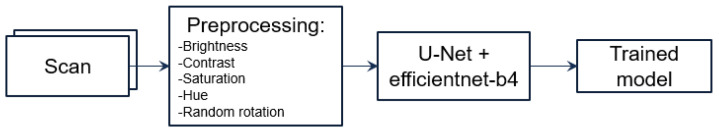
Training process of 2D model.

**Figure 5 sensors-24-00721-f005:**
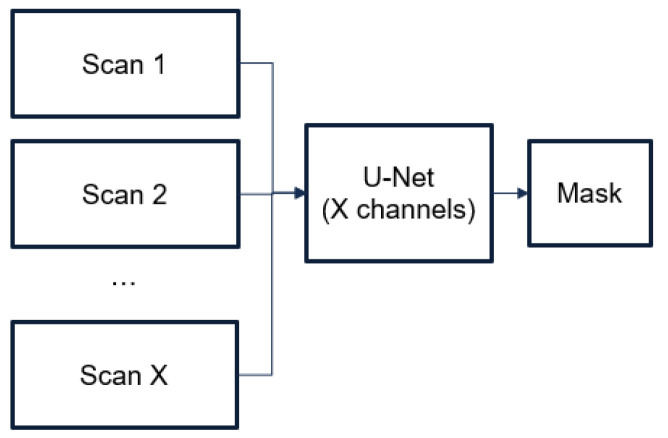
Architecture of 2.5D model.

**Figure 6 sensors-24-00721-f006:**
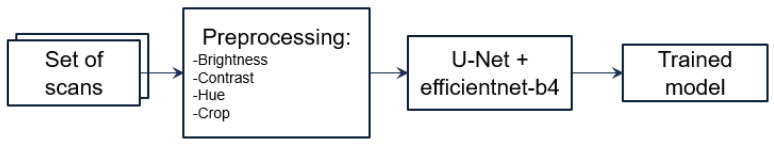
Training process of 2.5D model.

**Figure 7 sensors-24-00721-f007:**
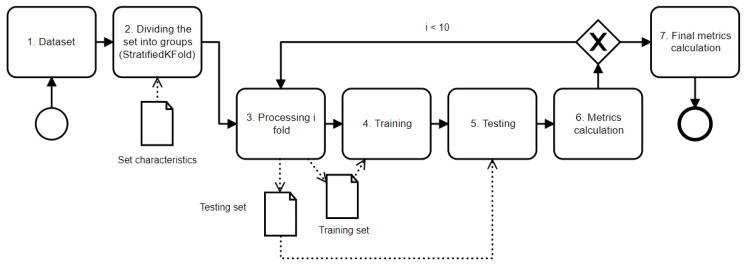
Scheme of cross-validation process (solid arrows show the relationship between stages and dotted arrows show data transfer between stages).

**Figure 8 sensors-24-00721-f008:**
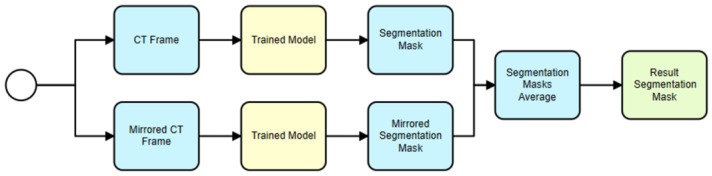
Trained model usage for hematoma segmentation (single U-Net model).

**Figure 9 sensors-24-00721-f009:**
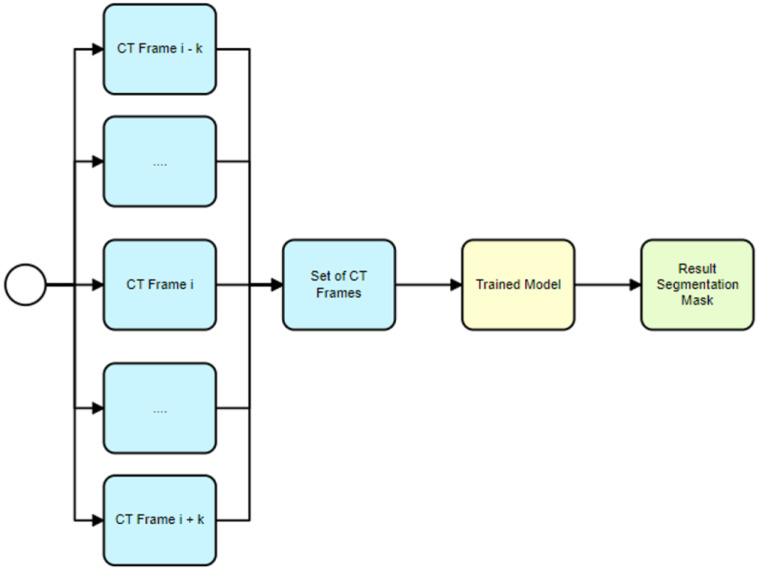
Trained model usage for hematoma segmentation (2.5D model).

**Table 1 sensors-24-00721-t001:** Characteristics of the collected dataset.

Parameter	Value
Number of patients	21
Minimum number of patients	1
Maximum number of patients	3
Overall number of series	53
Average number of scans for one series	≈60
Image size	512 × 512
Image format	DCOM
Color	Black and white
Overall dataset size, Gb	2.58
Series with one hematoma	40
Series with several hematomas	13
Series with embolization	26
Series without embolization	27

**Table 2 sensors-24-00721-t002:** Characteristics for 10-fold cross-validation set generation.

N	Embolization	Number of Hematomas
1	false	1
2	false	1
3	true	2
4	true	2
5	true	2
6	true	1
7	true	1
8	false	1
9	true	1
10	true	1
…	…	…
53	false	1

**Table 3 sensors-24-00721-t003:** Comparison of 2D model with different parameters (bold highlight the best results for different model parameters).

Model	Threshold	Jaccard (One Series)	Dice (One Series)	Jaccard (From Training)	Dice (From Training)	Jaccard (Mixed)	Dice (Mixed)
U-Net	0.3	0.5426	0.6505	0.9207	0.9582	0.6621	0.7835
U-Net	0.5	0.4282	0.5382	**0.9723**	**0.9858**	**0.6875**	**0.8002**
U-Net	0.7	0.4818	0.6091	0.948	0.9728	0.6821	0.799
U-Net (augmentation)	0.3	0.6281	0.7488	0.8659	0.9262	0.5513	0.6792
U-Net (augmentation)	0.5	**0.6537**	**0.7671**	0.8562	0.9186	0.5703	0.7073
U-Net (augmentation)	0.7	0.6281	0.753	0.9314	0.9641	0.6764	0.7951

**Table 4 sensors-24-00721-t004:** Comparison of 2.5D model with different parameters (bold highlight the best results for different model parameters).

Model	Threshold	Jaccard	Dice
U-Net(2.5D) (1 slice) (cross-valid)	0.1	0.6665	0.7754
U-Net(2.5D) (1 slice) (cross-valid)	0.2	0.6664	0.7754
U-Net(2.5D) (1 slice) (cross-valid)	0.3	0.6663	0.7754
U-Net(2.5D) (1 slice) (cross-valid)	0.5	0.6662	0.7754
U-Net(2.5D) (1 slice) (cross-valid)	0.7	0.6660	0.7754
U-Net(2.5D) (3 slices) (cross-valid)	0.1	**0.67**	**0.7798**
U-Net(2.5D) (3 slices) (cross-valid)	0.2	**0.67**	**0.7798**
U-Net(2.5D) (3 slices) (cross-valid)	0.3	0.6699	0.7798
U-Net(2.5D) (3 slices) (cross-valid)	0.5	0.6697	0.7798
U-Net(2.5D) (3 slices) (cross-valid)	0.7	0.6695	0.7798
U-Net(2.5D) (5 slices) (cross-valid)	0.1	0.6651	0.775
U-Net(2.5D) (5 slices) (cross-valid)	0.2	0.6651	0.775
U-Net(2.5D) (5 slices) (cross-valid)	0.3	0.6650	0.7750
U-Net(2.5D) (5 slices) (cross-valid)	0.5	0.6648	0.7750
U-Net(2.5D) (5 slices) (cross-valid)	0.7	0.6646	0.7750

**Table 5 sensors-24-00721-t005:** Results of clinical acceptance process.

Patient #, Age (Years), Gender	Location of Hematoma	Time Taken to Produce a Report (Min)							
		Manual Scenario				Automated Scenario			
		Exp#1	Exp#2	Exp#3	Mean	Exp#1	Exp#2	Exp#3	Mean
1, 77, male	Right unilateral	82	90	62	78	5.5	6	6	5.83
2, 82, male	Bilateral	132	141	94	122	17	25	20	20.67
3, 58, male	Left unilateral	36	52	44	44	9	14	10	11
4, 44, female	Bilateral	175	183	154	171	12	16	15	14.33
5, 60, male	Left unilateral	45	50	27	41	8.5	12	10	10.16

## Data Availability

Data are contained within the article.

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
