# Peer review of "AI-Based Approach to One-Click Chronic Subdural Hematoma Segmentation Using Computed Tomography Images"

_sensors, 2024, doi:10.3390/s24030721_

Round 1

Reviewer 1 Report

Comments and Suggestions for Authors

The authors present an approach for Hematoma Segmentation of Computer Tomography Images.  

The current presentation of the article lacks the novel aspects and the main findings of research.

The title of the paper is “Intelligent Approach to Human Health Monitoring Automation Based on Chronic Subdural Hematoma Segmentation of Computer Tomography Images” However the health monitoring part is not covered in any part of the paper.

The paper is poorly written, the language and the presentation need to be improved.

The contribution of the paper needs to be stated clearly.

The abstract should cover the work done so far and the problem that the authors try to solve. It would be feasible to present the data in percentages rather than 2–20 per 100,000 people. Some sentences in the abstract e.g., “The first model analyses one scan while the second model analyses several scans simultaneously that more similar to doctor behavior” as well as in the rest of the article are difficult to interpret.

In the abstract, the authors state that 53 research of 21 patients were used. (I think they mean images from the research). In the last paragraph of the introduction section, they state that 54 CT scans contain 21 people. In line 219 the statement is “Our dataset contains of 53 patients over 18 years old with the presence of CSDH”. In Table 1 number of patients is 21 and the overall number of patients is 53. What is the actual number of patients and the number of scans used need to be clearly stated with the terminologies used in the literature.

The focus of the paper has been on generating the dataset. The authors state that all requirements related to the regulations and patient privacy were approved before the study was conducted. I could not find the link to the generated dataset. The authors need to publish their dataset so that other researchers can use the data.

The methodology should cover the parameters and other related information in relation to the proposed method rather generic discussion on machine learning models.

The term “Number of research” in the last four rows of the table needs to be elaborated.

What is the significance of the results obtained? What do these results contribute to the scientific community? The validity of the results needs to be proved with comparative analysis.

Comments on the Quality of English Language

Overall, the language is weak. Some examples are referred in the comments.

Author Response

We thank the reviewer for valuable comments, we summarized it in the table and provide the detailed answers in the attached document.

Reviewer 2 Report

Comments and Suggestions for Authors

We analyzed a very interesting article that addresses a very important topic. Congratulations to the authors.  The paper presents an adequate structure (an "approach" section that could be Materials and Methods) and is supported by a set of recent bibliographical references relevant to the work. However, we identified some aspects that deserve the attention of the authors

- Line 17: The objectives of the work presented must appear after the brief introduction and framework of the work,

- Line 22: The authors mention “53 research of 21 patients” . 53 studies carried out on 21 patients? No?

The materials and methods/approach, results and main conclusions must be identified and well delimited.

The Abstract deserves a major revision

- Line 62: The expression “health monitoring automation” is very broad... and difficult to interpret in the context of the work presented,

- Line 65: The results should not appear at the end of the “Introduction” section. A paragraph should appear presenting what is covered in the different sections of the work that follow the Introduction.

In the “approach” section, the technical factors involved in carrying out CT studies are not mentioned, namely the slice thickness of the images and whether the acquisitions were helical or incremental. This aspect is very important and must be mentioned.

- Line 183: The segmentation was done only by a professional. Isn't this a limitation of the study?

- In lines 21 and 22 it is written: “our own 21 dataset that includes 53 investigation (studies????) of 21 patients with one and two hematomas”. In line 219 the authors mention “Our dataset contains of 53 patients over 18 years”. We do not know which was the real sample used in the work.

Other considerations:

- The tables should appear in a more appropriate format,

In conclusions. The sample is very small. This aspect should appear as a limitation of the study.

- Page 387: The authors mention that “experts estimated neural network segmentation as acceptable for clinical usage”. This conclusion is not supported by the work performed

A major revision is suggested

Author Response

(The authors gave the same response as above.)

Round 2

Reviewer 1 Report

Comments and Suggestions for Authors

Most of the comments in the previous revision are addressed. 

Reviewer 2 Report

Comments and Suggestions for Authors

Congratulations. The paper meets minimum requirements for publication